# Precise Regulation of the TAA1/TAR-YUCCA Auxin Biosynthesis Pathway in Plants

**DOI:** 10.3390/ijms24108514

**Published:** 2023-05-10

**Authors:** Pan Luo, Dong-Wei Di

**Affiliations:** 1College of Life Science and Technology, Gansu Agricultural University, Lanzhou 730070, China; 2State Key Laboratory of Soil and Sustainable Agriculture, Institute of Soil Science, Chinese Academy of Sciences, Nanjing 210008, China

**Keywords:** IPA pathway, transcriptional regulation, protein modification, feedback regulation, regulatory mechanism

## Abstract

The indole-3-pyruvic acid (IPA) pathway is the main auxin biosynthesis pathway in the plant kingdom. Local control of auxin biosynthesis through this pathway regulates plant growth and development and the responses to biotic and abiotic stresses. During the past decades, genetic, physiological, biochemical, and molecular studies have greatly advanced our understanding of tryptophan-dependent auxin biosynthesis. The IPA pathway includes two steps: Trp is converted to IPA by TRYPTOPHAN AMINOTRANSFERASE OF ARABIDOPSIS/TRYPTOPHAN AMINOTRANSFERASE RELATED PROTEINs (TAA1/TARs), and then IPA is converted to IAA by the flavin monooxygenases (YUCCAs). The IPA pathway is regulated at multiple levels, including transcriptional and post-transcriptional regulation, protein modification, and feedback regulation, resulting in changes in gene transcription, enzyme activity and protein localization. Ongoing research indicates that tissue-specific DNA methylation and miRNA-directed regulation of transcription factors may also play key roles in the precise regulation of IPA-dependent auxin biosynthesis in plants. This review will mainly summarize the regulatory mechanisms of the IPA pathway and address the many unresolved questions regarding this auxin biosynthesis pathway in plants.

## 1. Introduction

Auxin plays a vital role in regulating plant growth, development, and response to environmental stress [1,2,3,4]. Maintaining appropriate concentrations of free indole-3-acetic acid (IAA) is essential for the regulation of normal plant growth and development and for coping with biotic and abiotic stressors. Plants can maintain auxin homeostasis by regulating IAA biosynthesis, metabolism, and transport in vivo [5].

In plants, IAA is mainly synthesized through two pathways, the Trp-dependent and Trp-independent pathways [6]. The Trp-dependent pathway is further divided into four pathways depending on the different intermediate metabolites derived from Trp: the indole-3-pyruvic acid (IPA) pathway, the indole-3-acetamide (IAM) pathway, the tryptamine (TAM) pathway, and the indole-3-acetaldoxime (IAOx) pathway [6,7]. Among these pathways, the enzymes and biochemistry of the IPA pathway are best delineated.

In the IPA pathway, Trp is first converted into IPA by a reversible amino transfer reaction catalyzed by an enzyme in the TAA1/TARs family (Figure 1). The TAA1 gene was independently identified through mutant isolation by four research groups investigating shade avoidance [8], ethylene responses [9], responses to the auxin transport inhibitor N-1-napthylpthalamic (NPA) [10], and responses to cytokinin (CK) [11]. However, overexpression of *AtTAA1* exhibited no altered phenotypes, indicating that TAA1 encodes a key but not rate-limited enzyme [8,9,11]. The TAA1 protein belongs to a superfamily of pyridoxal-5′-phosphate (PLP)-dependent enzymes that have Trp aminotransferase activity [9,12]. The TAA1 protein uses L-Trp, but not D-Trp, as a substrate, as well as L-Phe, Tyr, Ala, Leu, Gln, and Met [13]. Genome-wide phylogenetic and functional analyses identified the *TAA1*/*TARs* genes in many species, including Arabidopsis, rice and maize (Appendix A) [8,9].

The IPA is then converted to IAA in a reaction mediated by a YUCCA-type flavin monooxygenase (FMO; Figure 1) [14,15]. *YUC* genes were first discovered through a genetic screen of activation-tagged lines in Arabidopsis. Gain-of-function mutants of *YUC1* (*yuc1D*) had high levels of auxin and auxin-induced phenotypes like epinastic cotyledons and long hypocotyls, which indicated that YUC genes encode a rate-limiting enzyme involved in auxin biosynthesis [16]. The *YUC* genes are functionally redundant, as single mutants of *YUC* genes in Arabidopsis exhibited wild-type-like phenotypes, except for *yuc8*/*ckrc2*, which exhibited root curling when grown on medium with exogenous cytokinin (CK) [17]. The first step in the YUC-catalyzed reaction is the reduction of the FAD cofactor by NADPH to FADH, which subsequently reacts with oxygen to form a flavin-C4a-(hydro)peroxide intermediate. Then, the C4a-hydroperoxyflavin reacts with IPA to produce IAA. In vitro, YUC6 can use either PPA or IPA as a substrate, suggesting that YUC enzymes do not have strict substrate specificity [18]. To date, members of the *YUC* gene family have been found in more than 20 species, including 11 genes in Arabidopsis, 14 genes in rice and 14 genes in maize (Appendix A) [19].

Genetic disruption of the IPA pathway, and the resulting dysregulation of IAA levels, leads to plant developmental defects under both normal and stress environments [19]. To maintain IAA homeostasis, plants have evolved multiple layers of regulatory mechanisms (Figure 1), including transcriptional regulation (layer I), post-transcriptional regulation (layer II), protein modification (layer III), and negative feedback regulation (layer IV). Transcriptional regulation mainly includes epigenetic modifications (DNA methylation and modification of histone in ribosomes) and transcription factor-mediated activation/repression of precursor-mRNA (pre-mRNA) synthesis. Immediate post-transcriptional regulation, including splicing, processing, storage, and stabilization of pre-mRNA, regulates the efficiency of mRNA translation into protein products that include truncated proteins. Finally, translated precursor proteins (pre-proteins) undergo a series of post-translational modifications (PTMs), such as phosphorylation, acetylation, ubiquitination and glycosylation, that alter the localization, stability, activity, and interaction of the protein with other proteins, ultimately determine the biological activity of the functional proteins. These regulatory processes are influenced not only by different environmental factors and hormonal signals, but also by feedback from both intermediate and final products, resulting in a complex and well-defined regulatory network. These controls form an elaborate regulatory network that collectively maintains the homeostasis of endogenous IAA (Figure 1) [1,6,19,20,21,22,23,24,25,26]. Biochemically, the enzymes in the IPA pathway can also be manipulated by synthetic chemical compounds. In this review, we systematically summarize the multi-level regulation of the IPA-dependent auxin biosynthesis pathway in plants.

## 2. Small Chemical Inhibitors Target TAA1/TARs and YUCCA to Modulate Auxin Synthesis

Due to the important role of IAA in plant growth and development, genes involved in IAA biosynthesis, metabolism, transport and signaling are often subject to tight genetic regulation. Auxin biosynthetic genes either show redundancy or their single mutants result in lethality or sterility, such that classical genetic approaches may not be able to comprehensively screen for key auxin-related genes. The use of small chemical inhibitors can complement classical genetics. These small molecules often competitively occupy the ligand binding pocket of the target enzymes and can be applied in discreet doses to give a wide range of effects [27,28,29]. To date, several auxin biosynthesis inhibitors have been found and widely used, including nalacin [30], NPA [31], and auxinole [32]. As the IPA pathway is by far the most well studied of the IAA biosynthesis pathways, the chemical synthesis inhibitors identified also focus on this pathway:

The compound L-kynurenine (Kyn) was found in a screen for ethylene (ET) signaling inhibitors. Exogenous application of Kyn results in root elongation that is insensitive to ET. Subsequent studies have shown that TAA1/TAR1 catalyzes the conversion of Kyn to kynurenic acid (KYNA), and that this metabolite has no inhibitory effect on root growth. Computational Docking and Molecular Modeling results further suggested that Kyn acts as a competitive inhibitor of Trp in TAA1/TAR proteins, thereby reducing conversion of IPA and decreasing the levels of free IAA [33]. Several other chemical inhibitors have been found to inhibit the activity of TAA1/TARs, including 2-amino-oxyisobutyric acid (AOIBA), Pyruvamine2031, L-aminooxy-phenylpropionic acid (AOPP), 2-(aminooxy)-3-(naphthalen-2-yl) propanoic acid (KOK1169/AONP), and the IPA analogs KOK2099 and KOK2052BP (Figure 2) [13,33,34,35,36]. There are also two compounds, amino ethoxyvinylglycine (AVG) and amino-oxyacetic acid (AOA), that more broadly inhibit the activities of PLP-dependent enzymes, including TAA1/TARs and 1-aminocyclopropane-1-carboxylic acid (ACC) synthase, in vivo [36].

A second class of IPA pathway inhibitors target the YUC proteins. Yucasin, or 5-(4-chlorophenyl)-4H-1,2,4-triazole-3-thiol, was identified from a screen for compounds affecting IAA contents in the coleoptile tip [37]. Yucasin shares a similar sub-structure with methimazole, which has been used as an artificial substrate for FMOs in vitro and is able to inhibit the function of yeast FMO [38]. Yucasin functions as a competitive inhibitor of recombinant AtYUC1, with a higher binding affinity than IPA, and inhibits YUC1 activity in a dose-dependent manner [39]. There are several other inhibitors of YUC activities, including 4-biphenylboronic acid (BBo), 4-phenoxyphenylboronic acid (PPBo), Yucasin DF and ponalrestat (Figure 2) [37,40,41].

## 3. Layer Ⅰ: Finely Tuned Transcriptional Regulation of IPA-Dependent Auxin Biosynthesis

### 3.1. Epigenetic Modification of Genes Involved in IPA-Dependent Auxin Biosynthesis Pathway

Epigenetic modifications, including DNA methylation and histone modification in nucleosomes, are critical layers of transcriptional regulation, directing mRNA synthesis and determining gene expression or silencing [25]. Several studies have focused on the roles of epigenetic modifications in IPA-dependent auxin biosynthesis.

In plants, DNA methylation is a reversible, yet relatively stable, conversion of a cytosine (C) base into a 5-methylcytosine, usually in a CG, C-(A/T/C)-G, or C-(A/T/C)-(A/T/C) sequence context, that most often results in gene silencing [22,42]. DNA methylation is introduced at a site when the DNA methyltransferase DOMAINS REARRANGED METHYLTRANSFERASE 1/2 (DRM1/2) catalyzes the methylation of DNA from two unmethylated strands, a process directed by a 24 nt small interfering RNA (siRNA) (also named the RdDM pathway) and is maintained at a site by METHYLTRANSFERASE 1 (MET1), CHROMOMETHYLASE 2 (CMT2) or CMT3 when a DNA strand is copied through semi-conservative replication of a methylated DNA [42,43]. Recent analysis of genome-wide methylation patterns has identified many genes in the IPA pathway (TAA1, TAR1/2, and YUC1/2/5/10) as targets of the RdDM pathway, suggesting that DNA methylation may play an important role in regulating the IPA-dependent auxin biosynthesis pathway [44]. However, there are few studies on the regulation of IAA homeostasis through DNA methylation in response to stress or during development. During screening of small RNA in response to different ambient temperatures, a, 24 nt siRNA (Locus_77297) was identified that directs the methylation of the *YUC2* promoter in a temperature-dependent way, which then blocks the binding of the transcription factor NUCLEAR FACTOR-YA2 (NF-YA2) to the *YUC2* promoter [45].

In addition to DNA methylation, modification of histones within nucleosomes, including histone H3 methylation, acetylation, and histone H2B monoubiquitination, also influences the transcriptional activity of genes [25]. The role of nucleosomal histone modification in the regulation of IAA synthesis and metabolism has been systematically summarized in our recently published review (reviewed by [25]), so this paper only briefly summarizes the genes with known histone modifications and the processes that these modifications impact (Table 1).

While epistatic modifications seem to regulate the IPA-dependent auxin biosynthesis pathway in response to stress and development, there are few relevant detailed studies. Future studies must be undertaken on how different developmental stages and different stresses epistatically alter the transcription of genes involved in the IPA-dependent IAA biosynthesis pathway.

### 3.2. Complex Transcriptional Regulatory Mechanisms of the TAA1/TAR and YUCCA Genes

Developmental phenotypes of different single, double and multiple mutants of the TAA1/TAR and YUC genes show that the IPA-dependent auxin biosynthesis pathway is involved in almost all aspects of plant growth and development, including seed germination, embryo development, hypocotyl growth, and leaf development [1,6,19]. Moreover, many essential transcription factors (TFs) have been identified that regulate the transcription of *TAA1/TAR* and *YUC* genes to influence different stages of plant growth and development.

#### 3.2.1. Vegetative Stage

The vegetative stage includes seed germination and the juvenile and adult phases [63]. During seed germination, the distribution of auxin determines the adaxial–abaxial polarity and then formation of the cotyledon and leaf growth [64]. In Arabidopsis, a pair of TFs, KANADI 1 (KAN1) and REVOLUTA (REV), play opposite roles in auxin distribution by directly binding to the promoters of *TAA1* and *YUC5*, with KAN1 repressing and REV promoting their transcription [64]. Together with the regulation of auxin transport (mediated by LAX2 and LAX3), the antagonistic function of KAN1 and REV result in maximum auxin levels at the site of cotyledon growth (Figure 3) [64]. In addition, two basic helix-loop-helix proteins, TARGET OF MONOPTEROS5 (TMO5)/TMO5-LIKE1 (T5L1) and LONESOME HIGHWAY (LHW), form a heterodimer complex and bind to the promoter of *YUC4*, leading to auxin accumulation during vascular cell development in the embryo [65]. Conversely, the IAA further promotes the transcription of LHW and TMO5/T5L1, indicating that there is a positive feedback regulation that fine-tunes the LHW-TMO5/T5L1 level during vascular development [65]. In rice, BABY BOOM 1 (BBM1) directly targets *OsYUC6*/*7*/*9* to prompt auxin biosynthesis, leading to somatic embryogenesis [66].

In the hypocotyl, the PIF4-*YUC8* regulatory module plays an important role in response to stress signals, including circadian rhythms, light, high temperature, and mechanical stress. The accumulation and transcriptional activity of PIF4 is regulated by different proteins, with competition for and interference at the YUC8 promoter by other transcription factors affect the positive regulation of *YUC8* by PIF4 and, consequently, the biosynthesis of auxin (Figure 3). In response to light, PIF4 interaction with PhyB results in the phosphorylation and then ubiquitination of PIF4, which is then degraded [67]. Another two TFs, DE-ETIOLATED 1 (DET1) and CONSTITUTIVE PHOTOMORPHOGENESIS 1 (COP1), promote high-temperature-induced hypocotyl growth by stabilizing PIF4 [68]. SEUSS (SEU) interacts with PIF4 and increases its binding and transcriptional activation activity in response to light and/or high temperature, while the interaction with CRY1 result in repression of PIF4 transcriptional activity under high temperature in a blue-light-dependent manner [69,70]. TIMING OF CAB EXPRESSION 1 (TOC1) accumulates more during evening and can repress activation the *YUC8* by PIF4 [71]. FLOWERING CONTROL LOCUS A (FCA) interacts with PIF4 and promotes PIF4 dissociation from the promoter of *YUC8*, attenuating PIF4 transcriptional activity under high temperature. PHYTOCHROME RAPIDLY REGULATED 1 (PAR1) interacts with PIF4 and inhibits its transcriptional activity in response to light signals. EARLY FLOWERING 3 (ELF3) interacts with PIF4 to prevent PIF4 from activating *YUC8*, while the accumulation of ELF3 is further regulated by phyB and COP1 in the light. LONG HYPOCOTYL IN FR LIGHT 1 (HFR1) interacts with PIF4 to form non-DNA-binding heterodimers that limit PIF4 transcriptional activity in the shade. Moreover, ELONGATED HYPOCOTYL 5 (HY5) can regulate hypocotyl elongation at high temperatures by competing with PIF4 for binding to *YUC8* [68]. Gibberellin (GA) antagonistically interacts with light signals through degradation of DELLA proteins, which can directly bind to the DNA-recognition domain of PIF4 and then block its transcriptional activity (Figure 3) [72]. In addition, the DELLA protein GAI interacts with ARABIDOPSIS RESPONSE REGULATOR 1 (ARR1) and enhances its transcriptional regulation of *TAA1* to regulate primary root growth [73]. Furthermore, PIF7 can directly bind to the *YUC8* promoter and form a heterodimer with PIF4 under high temperature [74].

In addition, another MYB-like transcription factor, REVEILLE 1 (REV1), is also involved in regulating hypocotyl growth by integrating YUC8-dependent auxin biosynthesis and circadian clock via a PIF4-independent pathway [75]. HOOKLESS 1 (HLS1) interacts with PIF4 to co-bind downstream gene promoters, including *YUC8*, in response to high temperature. Moreover, HLS1 is reported to respond to mechanical stress in an EIN3-dependent manner during soil emergence of seedlings [76]. It would be interesting to investigate whether the PIF4-*YUC8* module is also involved in this response. Additionally, some TFs, such as ZEITLUPE (ZTL) and MYB hypocotyl elongation-related (MYBH), have been reported to upregulate PIF4 transcription and to promote YUC8-dependent auxin biosynthesis; however, whether they act by directly binding to the PIF4 promoter remains unknown [77,78]. Taken together, these results indicate that the complex and finely tuned transcriptional regulation of *YUC8* is essential for maintaining hypocotyl growth in response to the environment.

Developmental signals activate another transcriptional pathway, the miR319-TCP4-*YUC5* module, to maintain cell expansion of the hypocotyl (Figure 3) [79]. Therefore, it would be interesting to investigate how stress signals and developmental signals synergistically regulate hypocotyl elongation in the future.

During root growth and development, the IPA-dependent pathway also plays an important role in integrating environmental stress and hormone signaling. For instance, jasmonic acid (JA) can promote lateral root development through the direct regulation of *YUC2* by ERF109 [80]. JA also employs a group of MYC TFs, MYC2/3/4, in response to mechanical wounding via directly activating *YUC8/9*-dependent auxin biosynthesis [81]. CK promotes auxin biosynthesis in roots, via ARR1 activation of *TAA1* transcription, while ARR12 synergically activates *TAA1* transcription via interaction with ARR1 [73]. Moreover, ET insensitive 3 (EIN3) is also involved in regulating the transcription of *TAA1* via direct interact with ARR1, leading to enhanced transcriptional activity of ARR1 [73]. In addition to *TAA1*, EIN3 also regulates *YUC5/8/9* in response to aluminum (Al) stress. Al stress promotes ET accumulation in the transition zone (TZ) of roots, and then activates two transcriptional pathways, namely EIN3-*YUC9* and EIN3-PIF4-*YUC5/8/9*, to promote auxin biosynthesis, resulting in inhibition of primary root growth under Al stress [82]. Furthermore, IAA promotes EIN3 accumulation in the nucleus via inhibiting EBF1/2 [33]. In rice, the homolog of EIN3, OsEIL1, is also involved in regulating ET-induced PR growth inhibition via directly activating the transcription of *OsYUC8* and *OsTAR2/MHZ10* [83,84]. Interestingly, two groups of Aux/IAA proteins, OsIAA1/9 and OsIAA21/31, can physically interact with OsEIL1 to promote and inhibit the activation of OsTAR2 by OsEIL1. ET treatment promotes degradation of the repressors IAA21/31 earlier than the activators IAA1/9 in a TIR1/AFB-dependent manner, leading to the activation of OsTAR2 by OsEIL1 [84]. Moreover, *OsYUC8* is also direct regulated by OsbZIP46 in primary roots during response to exogenous abscisic acid (ABA) [85]. Additionally, two homologous B3 TFs, FUSCA 3 (FUS3) and LEAF COTYLONDON 2 (LEC2), interact to bind to and activate *YUC4* during lateral root formation, while LEC2 also activate FUS3 transcription in lateral root initiation (Figure 3) [86].

In addition to these TFs, several others are also involved in regulating IAA levels in roots, although they have not been shown to directly regulate the TAA1/TAR1-YUC genes. For example, ABA can inhibit the transcription of *YUC2/8* via ABI4, thereby inhibiting primary root elongation. Mechanical wounding can upregulate ERF115, thereby promoting the transcription of *YUC3/5/7/8/9* and promoting post-injury root regeneration. ATH2 inhibits the transcription of *YUC2* to alter root gravitropism. AGL21 positively regulates *YUC5/8/TAR3*, and this TF is induced by a variety of hormones including IAA/ABA/JA and a variety of stresses, including salt and drought stress and sulfate (-S) and nitrogen deficiency (-N) (Figure 3) [87]. In conclusion, the transcriptional regulation of the IPA-dependent pathway in the root system plays an important role in coordinating root growth, hormonal signaling and stress response.

For leaf growth, NF-YA2 and NF-YA10 bind to and inhibit *YUC2*, which in turn decreases auxin content and leaf size [45]. Moreover, the miRNA miR169d targets these two TFs and cleaves them to maintain auxin biosynthesis during leaf growth (Figure 3) [45]. In addition, ARR1/10/12, which are involved in the regulation of shoot stem cell development through direct activation of *WUSCHEL* (WUS), also bind to the *YUC1/4* promoter, repressing *YUC1/4* transcription and indirectly promoting the induction of *WUS* by CK (Figure 3) [88].

#### 3.2.2. Reproductive Stage

Flower bud differentiation is a marker of the change from vegetative plant growth to reproductive growth [63]. During this stage, many *TAA1/TAR* and *YUC* genes are reported to regulate lateral organ morphogenesis and flower and seed development. Three INDETERMINATE DOMAIN (IDD) transcription factors, IDD14, IDD15, and IDD16, directly target *YUC5* and *TAA1* to promote auxin biosynthesis [89]. Overexpression or knockout of these *IDDs* result in pleiotropic phenotypes, including altered leaf shape, floral development and fertility, which can be repressed by mutation or overexpression of *YUC* genes, indicating the critical role of IPA-dependent auxin biosynthesis during the reproductive stage [89]. Another TF, SHORT-INTERNODES/STYLISH 1 (SHI/STY1) is also involved in regulating leaf and flower development via directly activating *YUC4* and indirect upregulating *YUC8* [90]. GROWTH REGULATING FACTOR 6 (GRF6) directly activates OsYUC1 and auxin biosynthesis during floral development, thus leading to increased branch and spikelet numbers [91]. GRF6 is further regulated by Os-miR396b, while blocking miR396b results in reshaping inflorescence architecture and increasing rice yield [91].

In addition to these TFs, which are useful for all organs at the reproductive growth stage, several tissue-specific TFs control local auxin biosynthesis and thus affect flower and seed development. For instance, SPATULA (SPT) integrates CK and auxin signaling via directly targeting *TAA1* in the medial domain of the gynoecium, and mutation of *SPT* leads to severe gynoecial developmental defects [92]. FT-INTERACTING PROTEIN 7 (FTIP7), highly expressed in anthers before mitotic division of pollen, facilitates nucleocytoplasmic translocation of the TF ORYZA SATIVA HOMEOBOX 1 (OSH1), which directly represses *OsYUC4* transcription and auxin biosynthesis during pollen mitosis, thus controlling the release of mature pollen (Figure 3) [93].

Furthermore, several TFs are involved in regulating seed development by directly regulating IPA pathway. For instance, LEAFY COTYLEDON 2 (LEC2) directly binds to the promoters of YUC2 and *YUC4* and activates their transcription, promotes somatic embryogenesis [94]. In rice endosperm, OsNF-YB1 binds to *OsYUC11* and activates its transcription, which is required for rice grain filling [95]. MATERNAL EFFECT EMBRYO ARREST 45 (MEE45) directly activates AINTEGUMENTA (ANT), and in turn ANT further activates the expression of *YUC4* in the ovule integument, resulting in embryo cell proliferation and determination of seed size [96]. ZmNF-YA13, a target of Zm-miR169o, directly induces the expression of *ZmYUC1* in early developing seeds, leading to a greater number of endosperm cells and a larger seed size (Figure 3) [97]. In addition to the TFs mentioned above in Arabidopsis, rice, and maize, several TFs have been reported to regulate *TAA/TAR* and *YUC* genes in other species (Appendix A).

## 4. Layer II: Post-Transcriptional Regulation of *TAA1/TAR* and *YUC* Genes in Plants

Post-transcriptional regulation of genes can affect the splicing, processing, storage and stability of mRNA, which in turn affects mRNA translation efficiency or the final product, such as creating truncated proteins [20]. Alternative splicing of *YUC4* results in the presence of two YUC4 isoforms, both of which have enzymatic activities in Arabidopsis. Of these splicing variants, YUCCA4.1 is present in all tissues and distributed throughout the cytoplasm, whereas YUCCA4.2 is present only in flowers and is localized to the cytoplasmic side of the endoplasmic reticulum membrane, which may confer properties related to subcellular compartmentation of IAA biosynthesis [98]. There is also alternative splicing of the IAA efflux transporters PIN-FORMED 4 (PIN4) and PIN7 [99,100]. In general, alternative splicing is detected in many genes involved in the IPA-dependent pathway, e.g., *TAR2*, *YUC2* and *YUC4*; however, how alternative splicing influences the expression of these genes needs further investigation.

Another form of RNA processing is polyadenylation, and its distribution in the 5′-untranslated region (UTR) and 3′-UTR is responsible for the stability of mature transcripts and influences their export to the cytoplasm, their subcellular localization, and recognition by the translational machinery [23,101]. A poly(A) tag sequencing approach showed that multiple alternative polyadenylations were detected in TAA1/TAR and YUC genes; however, it remains unknown whether these alternative polyadenylations are involved in the post-transcriptional regulation of genes related to auxin biosynthesis [23].

## 5. Layer III: Precise Control of IPA-Dependent Auxin Biosynthesis through Post-Translational Protein Modification

Post-translational modifications, such as phosphorylation, acetylation, ubiquitination and glycosylation, can affect protein localization, stability, activity and interactions with other proteins, adding additional complexity and greater flexibility to regulation of metabolic functions [102]. However, there are fewer reports on the post-translational modifications of IAA biosynthesis-related enzymes than on the transcriptional and epistatic modification regulation of IAA biosynthetic genes. A recent study showed that the AtTAA1 is phosphorylated at Threonine 101 (T101). Whether T101 is phosphorylated or not determines whether TAA1 is in the active or inactive state. TRANS-MEMBRANE KINASE 4 (TMK4) interacts with and then phosphorylates TAA1, resulting in suppression of TAA1 activity [103]. In addition, we used the CKRC (cytokinin induced root curling) system to screen for auxin-deficient mutants, and identified a low-auxin mutant, *ckrc3-1*, that was prematurely terminated due to a G to A transition at position 731 of the auxiliary subunit (Naa25) of the Arabidopsis N-TERMINAL ACETYLTRANSFERASE NatB [104]. CKRC3 interacts with the NatB catalytic subunit Naa20 (NBC) to form an active NatB complex and catalyzes the N-terminal acetylation (NTA) of the second amino acid at the N-terminal end of the protein, which is Aspartic acid (Asp, D), Asparagine (Asn, N) or Glutamic acid (Glu, E). Additionally, our results further showed that the CKRC3-NBC complex can catalyze the NTA of YUC8 and increase its stability to maintain auxin biosynthesis [104].

With the development of proteomics, many more types of protein modifications are being identified and studied. Many phosphorylation, acetylation and glycosylation modification sites have been identified on TAA1/TAR and YUC proteins. Whether these modifications are involved in the regulation of IPA-dependent IAA biosynthesis and how they are altered with plant development and stress deserve further investigation [105,106].

## 6. Layer IV: Negative Feedback Regulation of IPA Pathway

Negative feedback regulation is an important mechanism for maintaining the homeostasis of enzymatic reactions. Suzuki et al. [107] found that exogenous application of the synthetic auxins 1-naphthaleneacetic acid (NAA) and 2,4-dichlorophenoxyacetic acid (2,4-D) decreases the transcription of *TAR2*, *YUC1*, *YUC2*, *YUC4*, and *YUC6* in Arabidopsis seedlings, while use of the auxin biosynthetic inhibitor Kyn upregulated the transcription of these genes (Figure 4). Consistently, similar regulation was also observed in mutants with high or low endogenous IAA. These results suggested that the genes involved in the IPA pathway are transcriptionally regulated by negative feedback from active IAA levels [107].

Additionally, the product IPA can negatively regulate the activity of TAA1/TARs, through reversibility of the Trp aminotransferase activity and competitive inhibition of the TAA1/TARs by IPA (Figure 4). Other aminotransferases can catalyze reversible reactions; however, is remains unknown if the TAA1/TARs have this ability [108]. A recent study showed that IPA was converted to Trp in the presence of TAA1, but not heat-inactivated TAA1, suggesting that TAA1 also possesses reversible Trp aminotransferase activity, although this activity is much lower [13]. The IPA analog KOK2099 also inhibits the aminotransferase activity of TAA1, leading to a decrease in the endogenous IAA levels, while AtTAA1 activity was enhanced when the reaction mixture contained AtYUC10. These data suggested that KOK2099 and IPA strongly inhibit TAA1 activity (Figure 2 and Figure 4). Further investigation suggested that KOK2099 and IPA could mimic Trp and enter the active site of TAA1 (E-PLP); however, they could not form a Schiff base with TAA1 due to the lack of an amino moiety [13]. In addition, high concentrations of IPA were reported to inhibit recombinant AtYUC1 activity in vitro, indicating that feed-forward inhibition may also function in maintaining IPA homeostasis (Figure 4) [39]. Taken together, the negative feedback regulation of TAA1 ensures that plants do not accumulate too much IPA, thus maintaining IPA homeostasis. These feedback mechanisms are likely a key reason for which overexpression of TAA1 does not lead to excessive IAA accumulation [8,9,11,17].

Another way that the level of IPA is steadily maintained is the conversion of IPA Trp by REVERSAL OF SAV 1 (VAS1), which uses methionine as an amino donor and IPA as an amino acceptor to produce L-Trp and 2-oxo-4-methylthiobutyric acid. IPA can also be glucosylated into IPA-Glc by UGT76F1 (Figure 4) [109,110].

## 7. Concluding Remarks

Auxin is an essential hormone that governs plant development and responses to bio- or abiotic stress [1,111]. Study of the auxin biosynthetic pathways and their regulation at different layers is extremely important for both plant science and agricultural development. In addition, local auxin biosynthesis and distribution play essential roles in many developmental processes and stress responses [6,26,112]. However, many questions remain, particularly those surrounding regulation by DNA methylation and miRNAs.

Tissue-specific DNA methylation may regulate local IAA biosynthesis. Local auxin biosynthesis plays a critical role in the formation of the auxin gradient, which functions in regulating plant development and stress response [26]. Multiple copies of YUC genes in the plant genome may show tissue-specific expression, regulating local IAA biosynthesis [26]. However, the mechanism by which plants select one or a few YUCs for IAA synthesis at a specific location remains unclear. A recent study showed that in the *drm1drm2cmt3* triple mutant, which has low levels of DNA methylation, *YUC2* and *TAA1* were specifically induced in the leaves, but almost none was detected in the roots [113], implying that DNA methylation may be involved in the regulation of local IAA biosynthesis. In the future, studies on tissue-specific DNA methylation will provide insight into how plants regulate local IAA biosynthesis.

Silencing of transcription factors by miRNA may also influence local auxin biosynthesis. As short, single-stranded nucleic acids, miRNA directly cleave target genes and repress the expression, which provides an additional layer of regulation to gene expression [112]. Published studies showed that miRNAs and TFs may form a regulatory module to control *YUC* gene expression in specific tissues, leading to spatiotemporal auxin signaling [91,97,114]. Therefore, it is extremely important to discover tissue-specific miRNA-TFs regulatory modules and to explore the mechanisms of tissue-specific distribution of miRNAs, which will help to elucidate the molecular mechanisms of IPA-dependent local auxin biosynthesis.

Many studies have shown that auxins play a key regulatory role in enhancing plant stress resistance and improving crop yields [115,116,117]. However, modification of a specific functional gene (auxin-related) or exogenous auxin application has not achieved the desired effect [118]. This is due to the facts that: auxin homeostasis is controlled at the levels of biosynthesis, metabolism, degradation and transport, and that auxin tends to act only on a specific tissue, or even a specific region of a tissue, and indiscriminately changing auxin levels in the whole plant can have unpredictable effects on overall growth [26]. In view of this, we need to explore more tissue-specific or even region-specific promoters to alter the auxin signal in a particular region to develop finer gene editing techniques to accomplish site-specific gene editing.

## Figures and Tables

**Figure 1 ijms-24-08514-f001:**
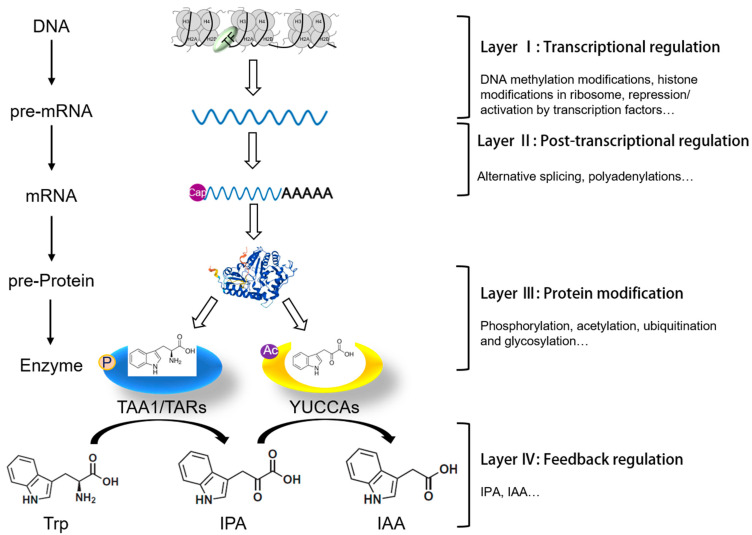
Overview of IPA-dependent pathway regulation. Auxin biosynthesis through the IPA pathway is controlled through multiple layers of regulation. The first layer, transcriptional regulation, includes DNA methylation, histone modification in ribosome, repression/activation by transcription factors. The second layer, post-transcriptional regulation, includes alternative splicing and polyadenylation. The third layer is protein modification, which includes phosphorylation, acetylation, ubiquitination and so on. The fourth layer is feedback regulation of gene transcription and enzyme activities of TAA1/TARs and YUCs induced by accumulation of IPA and/or IAA.

**Figure 2 ijms-24-08514-f002:**
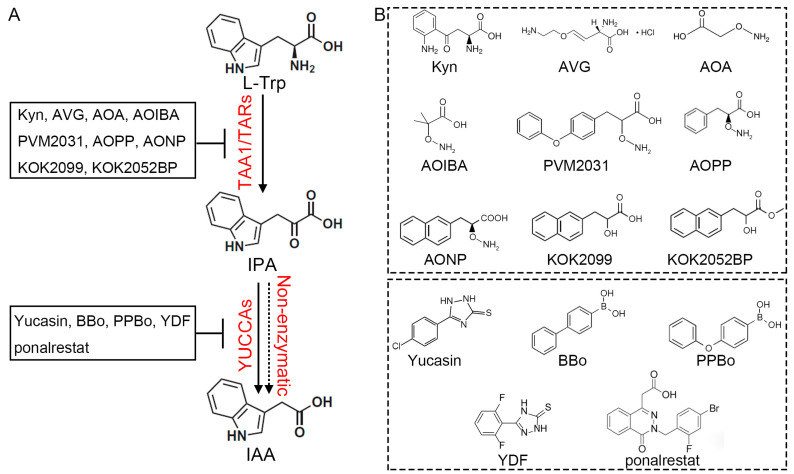
The enzymes and chemical inhibitors involved in the IPA-dependent auxin biosynthesis pathway. (**A**) The enzymes involved in IPA-dependent auxin biosynthesis; (**B**) the chemical structures of auxin biosynthetic inhibitors. L-kynurenine, Kyn; 2-amino-oxyisobutyric acid, AOIBA; Pyruvamine2031, PVM2031; L-aminooxy-phenylpropionic acid, AOPP; 2-(aminooxy)-3-(naphthalen-2-yl) propanoic acid, AONP; amino ethoxyvinylglycine, AVG; amino-oxyacetic acid, AOA; 5-(4-chlorophenyl)-4H-1,2,4-triazole-3-thiol, Yucasin; 4-biphenylboronic acid, BBo; 4-phenoxyphenylboronic acid, PPBo.

**Figure 3 ijms-24-08514-f003:**
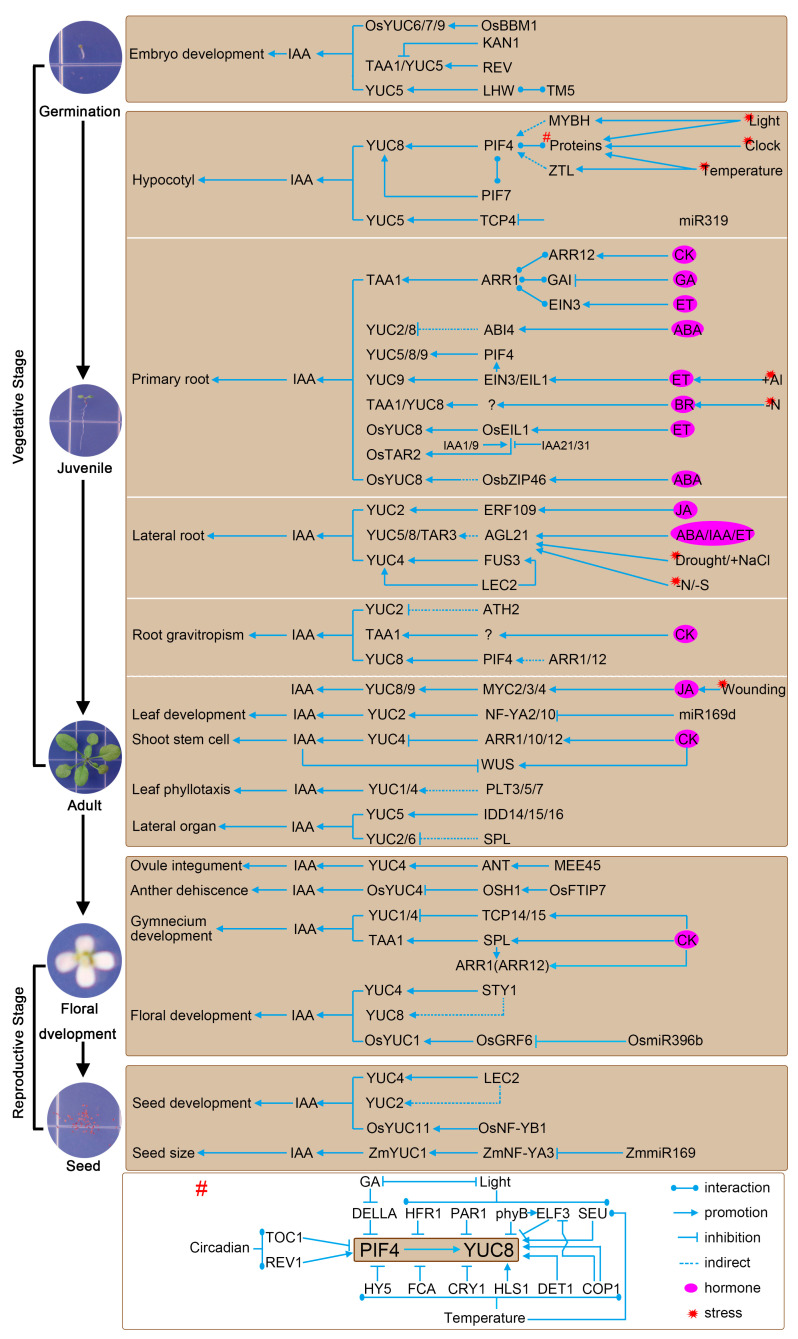
Transcriptional regulation of TAA1/TAR and YUC genes. # indicates details of PIF4 and its interacting proteins.

**Figure 4 ijms-24-08514-f004:**
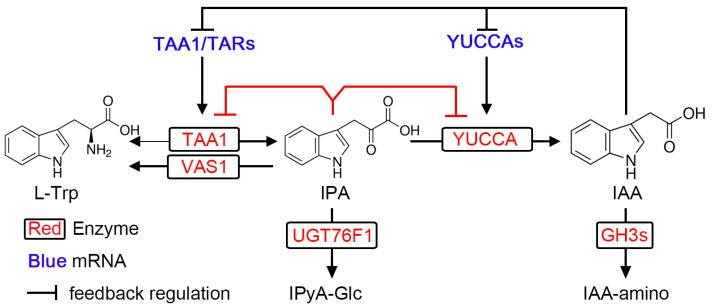
Negative feedback regulation of IPA-dependent auxin biosynthesis pathway.

**Table 1 ijms-24-08514-t001:** Epigenetic modifications of YUC genes.

Gene	Regulated by	Related Developmental Process	Ref.
*YUC1*	SUP-LHP1-PRC2 complex	Floral patterning	[46]
*YUC2*	DRM1 and DRM2	Leaf growth and thermomorphogenesis	[45,47]
*YUC3*	BRM and REF6	n.s.	[48]
*YUC4*	GCN5/HAG1	Gynoecium development	[49]
*YUC4*	SUP-LHP1-PRC2 complex	Floral whorl boundaries	[46]
*YUC4*	CLF, LHP1, CHR11, and CHR17	Floral patterning and floral determinacy	[46,50]
*YUC6*	SWI3B	Leaf blade development	[51]
*YUC7*	HUB complex	Root gravitropism	[52]
*YUC8*	FCA	Thermal adaptation of stem growth	[53]
*YUC8*	PIF7-MRG2 complex	Shade-induced hypocotyl elongation	[54]
*YUC8*	SWR1 chromatin remodeling complex	Thermomorphogenesis	[55]
*YUC8*	HDA9-PWR complex	Thermomorphogenesis	[56]
*YUC8*	INO80 chromatin remodeling complex	Thermomorphogenesis	[57]
*YUC8*	JMJ14, JMJ15, and JMJ18	Response to high temperature	[58]
*YUC9*	ARP4	Shade-induced hypocotyl elongation	[59]
*YUC10*	FIS2-PRC2 complex	Endosperm development	[60]
*YUC10*	EML1 and EML3	Seed development	[61]
*YUCs*	TFL2/LHP1	n.s.	[62]
*YUCs*	JMJ12/REF6	Thermomorphogenesis	[58]

SUPERMAN, SUP; LIKE HETEROCHROMATIN 1, LHP1; Polycomb Repressive Complex 2, PRC2; DOMAINS REARRANGED METHYLTRANSFERASE 1/2, DRM1/2; BRAHMA, BRM; RELATIVE OF EARLY FLOWERING 6, REF6; GENERAL CONTROL NONREPRESSIBLE 5, GCN5; HISTONE ACETYLTRANSFERASE OF THE GNAT FAMILY 1, HAG1; CURLY LEAF, CLF; CHROMATIN REMODELING 11/17, CHR11/17; FLOWERING CONTROL LOCUS A, FCA; MORF-RELATED GENE 2, MRG2; SWI2/SNF2-RELATED 1, SWR1; HISTONE DEACETYLASE 9, HDA9; POWERDRESS, PWR; INOSITOL AUXOTROPHY 80, INO80; JUMONJI DOMAIN-CONTAINING 14/15/18, JMJ14/15/18; ACTIN-RELATED PROTEIN 4, ARP4; FERTILIZATION-INDEPENDENT SEED 2, FIS2; EMSY-Like protein 1/3, EML1/3; TERMINAL FLOWER 2, TFL2; n.s. stands for non-studied.

## Data Availability

Not applicable.

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
