# Peer review of "Precise Regulation of the TAA1/TAR-YUCCA Auxin Biosynthesis Pathway in Plants"

_ijms, 2023, doi:10.3390/ijms24108514_

Round 1

Reviewer 1 Report

In the manuscript, the authors summarized recent research progresses towards the understanding of the regulatory mechanisms of TAA1/TARs and YUCCAs, which are crucial genes in IPyA-dependent auxin biosynthesis pathway. The authors also provided insights into the transcriptional, epigenetic, and post-translational regulation of TAA1/TARs and YUCCAs that play important role in the maintenance and regulation of auxin homeostasis during plant development. The manuscript is well organized and the content is extensive. Before the manuscript could be accepted for publication, the authors need to address the following concerns.

Major comments:

1.     In the title and abstract, the authors mentioned “plant kingdom” multiple times, however, in later parts, the major discussion focused on the model plant Arabidopsis. The authors should include more functional studies of TAA1/TARs and YUCCAs in plants other than Arabidopsis. Although the authors did mention many aspects of the regulation of TAA1/TARs and YUCCAs in other species. 

2.     The authors should recheck the small molecule regulators of auxin homeostasis based on several recent review papers (10.1101/cshperspect.a040105; https://doi.org/10.3390/life12081285; https://doi.org/10.1080/09168451.2018.1462693). There is another inhibitor for YUC, ponalrestat (10.1074/jbc.RA119.010480).

3.     Language editing is recommended given the presence of multiple grammatical ambiguity, for example, the sentences on line 9, line 126, and line 153.

Minor comments: 

Line 34: “completed” should be “completely”

Line 48: “more and more species” should be “many species”

Line 60: the sentence “Then, the C4a intermediate with IPyA to produce IAA.” is confusing.

Line 62: “to data” should be “to date”?

Line 109-110: the sentence is ambiguous, I suppose the authors are trying to state that “These results suggested that the genes involved in IPyA pathway are transcriptionally regulated by a negative feedback from active IAA level”.

Line 174: “accumulation” should be “accumulates”

Line 219: “resulting to” should be “resulting in”

Line 282: “active” should be “activates”

Author Response

In the manuscript, the authors summarized recent research progresses towards the understanding of the regulatory mechanisms of TAA1/TARs and YUCCAs, which are crucial genes in IPyA-dependent auxin biosynthesis pathway. The authors also provided insights into the transcriptional, epigenetic, and post-translational regulation of TAA1/TARs and YUCCAs that play important role in the maintenance and regulation of auxin homeostasis during plant development. The manuscript is well organized and the content is extensive. Before the manuscript could be accepted for publication, the authors need to address the following concerns.

Major comments:

  1. In the title and abstract, the authors mentioned “plant kingdom” multiple times, however, in later parts, the major discussion focused on the model plant Arabidopsis. The authors should include more functional studies of TAA1/TARs and YUCCAs in plants other than Arabidopsis. Although the authors did mention many aspects of the regulation of TAA1/TARs and YUCCAs in other species. 

We have added more functional studies of TAA1/TARs and YUCCAs in other species and listed them in the new Supplementary Table 1.

  1. The authors should recheck the small molecule regulators of auxin homeostasis based on several recent review papers (10.1101/cshperspect.a040105; https://doi.org/10.3390/life12081285; https://doi.org/10.1080/09168451.2018.1462693). There is another inhibitor for YUC, ponalrestat (10.1074/jbc.RA119.010480).

We thank the reviewer for the suggestions, and we have read these papers carefully and discussed and cited them in the revised version. In addition, we also added the ponalrestat in the new Figure 2.

  1. Language editing is recommended given the presence of multiple grammatical ambiguity, for example, the sentences on line 9, line 126, and line 153.

 We have carefully considered your comments, and have tried our best to revise any grammatical errors in the new manuscript. We also invited a native English speaker to revise the manuscript carefully.

Minor comments: 

Line 34: “completed” should be “completely”

Done as suggested.

Line 48: “more and more species” should be “many species”

Done as suggested.

Line 60: the sentence “Then, the C4a intermediate with IPyA to produce IAA.” is confusing.

We have rewritten this sentence to say: “Then, the C4a-hydroperoxyflavin reacts with IPA to produce IAA.”.

Line 62: “to data” should be “to date”?

Done as suggested.

Line 109-110: the sentence is ambiguous, I suppose the authors are trying to state that “These results suggested that the genes involved in IPyA pathway are transcriptionally regulated by a negative feedback from active IAA level”.

We have followed this excellent advice, and have rewritten the sentence to now say “These results suggested that the genes involved in the IPA pathway are transcriptionally regulated by negative feedback from active IAA levels

Line 174: “accumulation” should be “accumulates”

Done as suggested.

Line 219: “resulting to” should be “resulting in”

Done as suggested.

Line 282: “active” should be “activates”

Done as suggested.

Reviewer 2 Report

 This manuscript presents important research on the biosynthesis of auxin, but there are areas that need improvement. The organization of the content related to auxin biosynthesis is not systematic and tends to be too detailed. The Introduction section lacks sufficient background information and does not lead logically into subsequent sections. The Abstract should provide a more specific summary of the authors' focus on TAA1/TARs-YUCCAs.

1.   The proper abbreviation for indole-3-pyruvic acid is IPA, which is more commonly used than IPyA. The misspelling in Line 9 may have been an oversight or a typographical error.

2.    It is true that the Abstract section could be improved by providing more specific details about the authors' focus on TAA1/TARs-YUCCAs. The abstract should provide a concise summary of the main findings and arguments presented in the manuscript.

3.   The Introduction section could benefit from more introductory content that provides context and background information on the topic. The logic development should be structured in a way that leads to the subsequent sections. The description of the discovery process of inhibitors may not be necessary in this section, but it could be included in the section where the functions of the inhibitors are discussed in more detail. As a review paper, the authors should aim to provide a comprehensive overview of the relevant research and discuss the implications of their findings.

4.    An explanation should be added that aminoethoxyvinylglycine (AVG) and amino-oxyacetic acid (AOA) are inhibitors of TAA1/TARs because they affect pyridoxal 5′-phosphate (PLP)-dependent enzymes.

5.    In Figure 2, the mRNA is written under the name of the enzyme, giving the impression that IAA is a direct transcriptional inhibitor of YUCCA and TAA1/TARs. Although there is an explanation related to transcription in Figure 3, there is no such description on page 3, which explains Figure 2. Therefore, it cannot be concluded that IAA is a transcriptional inhibitor.

6.    Although many contents related to auxin are provided, there is a lack of systematic organization. In particular, the authors describe the regulation of auxin biosynthesis in different layers, but it tends to be too detailed and covers too many layers. The authors should aim to provide a clear and concise explanation.

7.    It is necessary to restructure the manuscript by condensing it into the two main themes highlighted in the "7. Concluding Remarks" section: "tissue-specific DNA methylation" and "tissue-specific regulatory modules of miRNA-TFs." The authors should aim to provide a more focused and cohesive narrative that emphasizes these themes.

Author Response

This manuscript presents important research on the biosynthesis of auxin, but there are areas that need improvement. The organization of the content related to auxin biosynthesis is not systematic and tends to be too detailed. The Introduction section lacks sufficient background information and does not lead logically into subsequent sections. The Abstract should provide a more specific summary of the authors' focus on TAA1/TARs-YUCCAs.

We are very grateful to the reviewer for his/her constructive comments and have done our best to reorganize and revise the manuscript in line with the reviewer's comments.

  1.  The proper abbreviation for indole-3-pyruvic acid is IPA, which is more commonly used than IPyA. The misspelling in Line 9 may have been an oversight or a typographical error.

We thank the reviewer for the suggestion, and we have revised ‘IPyA’ to ‘IPA’ in revised manuscript and corrected the typographical error in Line 9.

  1.   It is true that the Abstract section could be improved by providing more specific details about the authors' focus on TAA1/TARs-YUCCAs. The abstract should provide a concise summary of the main findings and arguments presented in the manuscript.

Many thanks to the reviewer for his/her valuable suggestions, we have modified the Abstract to now say: “The indole-3-pyruvic acid (IPA) pathway is the main auxin biosynthesis pathway in the plant kingdom. Local control of auxin biosynthesis through this pathway regulates plant growth and development and the responses to biotic and abiotic stresses. During the past decades, genetic, physiological, biochemical, and molecular studies have greatly advanced our understanding of tryptophan-dependent auxin biosynthesis. The IPA pathway includes two steps: Trp is converted to IPA by TRYPTOPHAN AMINOTRANSFERASE OF ARABIDOPSIS/ TRYPTOPHAN AMINOTRANSFERASE RELATED PROTEINs (TAA1/TARs), and then IPA is converted to IAA by the flavin monooxygenases (YUCCAs). The IPA pathway is regulated at multiple levels, including transcriptional and post-transcriptional regulation, protein modification, and feedback regulation, resulting in changes in gene transcription, enzyme activity and protein localization. Ongoing research indicates that tissue-specific DNA methylation and miRNA-directed regulation of transcription factors may also play key roles in the precise regulation of IPA-dependent auxin biosynthesis in plants. This review will mainly summarize the regulatory mechanisms of the IPA pathway and address the many unresolved questions regarding this auxin biosynthesis pathway in plants.”

  1.  The Introduction section could benefit from more introductory content that provides context and background information on the topic. The logic development should be structured in a way that leads to the subsequent sections. The description of the discovery process of inhibitors may not be necessary in this section, but it could be included in the section where the functions of the inhibitors are discussed in more detail. As a review paper, the authors should aim to provide a comprehensive overview of the relevant research and discuss the implications of their findings.

Many thanks to the reviewer's suggestions, we have revised the introduction to make it more logical and conducive to the presentation and discussion of the followed contents. The Introduction is now described as: Genetic disruption of the IPA pathway, and the resulting dysregulation of IAA levels, leads to plant developmental defects under both normal and stress environments. To maintain IAA homeostasis, plants have evolved multiple layers of regulatory mechanisms (Fig. 1), including transcriptional regulation (layer I), post-transcriptional regulation (layer Ⅱ), protein modification (layer III), and negative feedback regulation (layer IV). Transcriptional regulation mainly includes epigenetic modifications (DNA methylation and modification of histone in ribosomes) and transcription factor-mediated activation/repression of precursor-mRNA (pre-mRNA) synthesis. Immediate post-transcriptional regulation, including splicing, processing, storage, and stabilization of pre-mRNA, regulates the efficiency of mRNA translation into protein products that include truncated proteins. Finally, translated precursor proteins (pre-proteins) undergo a series of post-translational modifications (PTMs), such as phosphorylation, acetylation, ubiquitination and glycosylation, that alter the localization, stability, activity, and interaction of the protein with other proteins, ultimately determine the biological activity of the functional proteins. These regulatory processes are influenced by not only different environmental factors and hormonal signals, but also by feedback from both intermediate and final products, resulting in a complex and well-defined regulatory network. These controls form an elaborate regulatory network that collectively maintains the homeostasis of endogenous IAA (Fig. 1). Biochemically, the enzymes in the IPA pathway can also be manipulated by synthetic chemical compounds. In this review, we systematically summarize the multi-level regulation of the IPA-dependent auxin biosynthesis pathway in plants.” In addition, we also added a new Figure 1 to overview the contents of manuscript.

  1.   An explanation should be added that aminoethoxyvinylglycine (AVG) and amino-oxyacetic acid (AOA) are inhibitors of TAA1/TARs because they affect pyridoxal 5′-phosphate (PLP)-dependent enzymes.

We thank the reviewer for the suggestion, and we have rewritten these two inhibitors in a new paragraph in the revised manuscript, which described as: “There are also two compounds, amino ethoxyvinylglycine (AVG) and amino-oxyacetic acid (AOA), that more broadly inhibit the activities of PLP-dependent enzymes, including TAA1/TARs and 1-aminocyclopropane-1-carboxylic acid (ACC) synthase, in vivo. 

  1.   In Figure 2, the mRNA is written under the name of the enzyme, giving the impression that IAA is a direct transcriptional inhibitor of YUCCA and TAA1/TARs. Although there is an explanation related to transcription in Figure 3, there is no such description on page 3, which explains Figure 2. Therefore, it cannot be concluded that IAA is a transcriptional inhibitor.

Thank you very much for your suggestion, I have recreated the Figure and added more explanatory information, please see new Figure 3 for details.

  1.   Although many contents related to auxin are provided, there is a lack of systematic organization. In particular, the authors describe the regulation of auxin biosynthesis in different layers, but it tends to be too detailed and covers too many layers. The authors should aim to provide a clear and concise explanation.

Thanks to your constructive comments, we have added more content to the Introduction section to make the manuscript clearer and more logical, the contents described as follows: “To maintain IAA homeostasis, plants have evolved multiple layers of regulatory mechanisms (Fig. 1), including transcriptional regulation (layer I), post-transcriptional regulation (layer Ⅱ), protein modification (layer III), and negative feedback regulation (layer IV). Transcriptional regulation mainly includes epigenetic modifications (DNA methylation and modification of histone in ribosomes) and transcription factor-mediated activation/repression of precursor-mRNA (pre-mRNA) synthesis. Immediate post-transcriptional regulation, including splicing, processing, storage, and stabilization of pre-mRNA, regulates the efficiency of mRNA translation into protein products that include truncated proteins. Finally, translated precursor proteins (pre-proteins) undergo a series of post-translational modifications (PTMs), such as phosphorylation, acetylation, ubiquitination and glycosylation, that alter the localization, stability, activity, and interaction of the protein with other proteins, ultimately determine the biological activity of the functional proteins. These regulatory processes are influenced by not only different environmental factors and hormonal signals, but also by feedback from both intermediate and final products, resulting in a complex and well-defined regulatory network. These controls form an elaborate regulatory network that collectively maintains the homeostasis of endogenous IAA (Fig. 1) [1, 6, 19-26]. Biochemically, the enzymes in the IPA pathway can also be manipulated by synthetic chemical compounds. In this review, we systematically summarize the multi-level regulation of the IPA-dependent auxin biosynthesis pathway in plants.”

In addition, we also added a new Figure 1 to overview the contents of manuscript.

  1.   It is necessary to restructure the manuscript by condensing it into the two main themes highlighted in the "7. Concluding Remarks" section: "tissue-specific DNA methylation" and "tissue-specific regulatory modules of miRNA-TFs." The authors should aim to provide a more focused and cohesive narrative that emphasizes these themes.

Thank you for your valuable suggestions. We have restructured the manuscript and included Figure 1 to enhance the clarity of the article's logical framework. However, research on the methylation and miRNA regulation of IAA tissue-specific biosynthesis remains limited. Although current studies suggest the potential involvement of miRNAs and methylation in IAA tissue-specific biosynthesis, systematic summaries and conclusive findings are yet to be established.

Reviewer 3 Report

Authors submitted a manuscript entitled “Precise Regulation of TAA1/TARs-YUCCAs Auxin Biosynthe-2 sis Pathway in Plant Kingdom” to IJMS. This manuscript describes the regulatory mechanisms of the IPyA pathway and explains the associated pathways to auxin biosynthesis and the key players involved in this mechanism. The review is well-written and covers most of the aspects. I have very few comments and suggestions.

L13-14; provide full names when any abbreviation appears first.

L24; IAA, write full form and abbreviate here. In the next section, only abbreviations can be used. See the whole manuscript and fix these issues.

L32-34; re-write this sentence. Keep one word confirmed or completed.

L62; To data>To date

Sec 3: Do environmental factors affect the negative feedback regulation of the IPyA pathway? Is there any regulatory element/s reported which can mimic negative feedback regulation of the IPyA pathway in plants?

Sec 4: Are there any post-transcriptional regulatory mechanisms that also influence the TAA1/TARs-YUCCAs pathway?

Author Response

Authors submitted a manuscript entitled “Precise Regulation of TAA1/TARs-YUCCAs Auxin Biosynthesis Pathway in Plant Kingdom” to IJMS. This manuscript describes the regulatory mechanisms of the IPyA pathway and explains the associated pathways to auxin biosynthesis and the key players involved in this mechanism. The review is well-written and covers most of the aspects. I have very few comments and suggestions.

L13-14; provide full names when any abbreviation appears first.

Done as suggested.

L24; IAA, write full form and abbreviate here. In the next section, only abbreviations can be used. See the whole manuscript and fix these issues.

Done as suggested.

L32-34; re-write this sentence. Keep one word confirmed or completed.

We have changed ' completed ' to 'completely' in the revised version.

L62; To data>To date

Done as suggested.

Sec 3: Do environmental factors affect the negative feedback regulation of the IPyA pathway? Is there any regulatory element/s reported which can mimic negative feedback regulation of the IPyA pathway in plants?

This is an intriguing question. As far as we know, no environmental factors or regulatory elements have been identified as being associated with the negative feedback regulation of the IPA pathway. We hypothesize that this may be attributed to the lack of discrimination in the study regarding the extent of the contribution of negative feedback regulation.

Sec 4: Are there any post-transcriptional regulatory mechanisms that also influence the TAA1/TARs-YUCCAs pathway?

We thank the reviewer for the suggestion, and we meticulously searched the literature for post-transcriptional regulation of TAA1 and YUC and added a new section, “Layer II: Neglected post-transcriptional regulation of TAA1/TARs-YUCCAs in plants”, to the revised manuscript.

Round 2

Reviewer 1 Report

The authors had made adequate responses to all my concerns. I have no more comment.